# EFFICIENT GENERATION OF STRUCTURED OBJECTS WITH CONSTRAINED ADVERSARIAL NETWORKS

## ABSTRACT

Despite their success, generative adversarial networks (GANs) cannot easily generate *structured objects* like molecules or game maps. The issue is that such objects must satisfy structural requirements (e.g., molecules must be chemically valid, game maps must guarantee reachability of the end goal) that are difficult to capture with examples alone. As a remedy, we propose constrained adversarial networks (CANs), which embed the constraints into the model during training by penalizing the generator whenever it outputs invalid structures. As in unconstrained GANs, new objects can be sampled straightforwardly from the generator, but in addition they satisfy the constraints with high probability. Our approach handles arbitrary logical constraints and leverages knowledge compilation techniques to efficiently evaluate the expected disagreement between the model and the constraints. This setup is further extended to hybrid logical-neural constraints for capturing complex requirements like graph reachability. An extensive empirical analysis on constrained images, molecules, and video game levels shows that CANs efficiently generate valid structures that are both high-quality and novel.

## 1 INTRODUCTION

Generative Adversarial Networks (GANs) (Goodfellow et al., 2014) have shown impressive performance on challenging tasks – like image generation (Karras et al., 2018), text-to-image (Zhang et al., 2017), and style transfer (Zhu et al., 2017) – where the goal is to produce believable configurations.

A number of important applications, however, require to generate objects that are *both* credible *and* feasible with respect to hard structural constraints. Examples include generating drug molecules, which must satisfy chemical validity requirements, and game levels, where the goal must be reachable from the starting position. Recent studies (Guimaraes et al., 2017; De Cao & Kipf, 2018; Xue & van Hoeve, 2019) observed that GANs struggle in these tasks. The main reason is that the training examples alone are insufficient to capture the feasibility constraints and thus to guide the model toward producing valid objects.

As a remedy, we propose Constrained Adversarial Networks (CANs), a class of generative models that extend GANs to structured domains. Given a set of examples drawn from a latent distribution and a set of structural constraints, CANs learn to output valid structured with high probability. This is achieved by augmenting the standard GAN loss with a penalty term that discourages the model from producing infeasible structures. Since CANs inject the constraints directly into the learned model, valid structures can be obtained with high probability by performing forward inference on the generator, avoiding the need for costly sampling or optimization steps (Volz et al., 2018).

The penalty term is implemented using the semantic loss (SL) (Xu et al., 2018). The SL leverages knowledge compilation (Darwiche, 2011) to represent arbitrary Boolean constraint as a circuit and uses the latter to measure the mass allocated by the generator to infeasible objects. The procedure is probabilistically sound, exact, and does not require sampling. Notably, the circuit, which can be quite large depending on the complexity of the constraints, can be thrown away after learning.

In addition, we show how to extend the SL complex constraints that would normally lead to intractably large circuits. This is accomplished by first using a neural network to map configurations to a different space in which the structural constraints can be compactly encoded, and then applying the SL to the latter. This enables us to deal with reachability on a graph, which is beyond the reach of

standard SL. Finally, we show how the validity constraints baked into the generator can be enabled or disabled during inference using ideas from InfoGANs Chen et al. (2016).

In summary, we contribute:

- Constrained Adversarial Networks (CANs), a generalization of GANs in which the generator is encouraged (at training time) to output valid structures with high probability via forward inference.
- An extensive empirical analysis on structured objects like constrained images, molecules, and video game levels showing that CANs generate structures that are stylistically coherent with the training data.
- A decomposition of intractably complex constraints into a neural and logical components, showcased on graph reachability for video game level synthesis.

## 2 Unconstrained GANs

Let $\mathcal{X}$ be the object space and $\mathcal{Z}$ some latent space. GANs (Goodfellow et al., 2014) are composed of a generator $g : \mathcal{Z} \to \mathcal{X}$ and a discriminator $d : \mathcal{X} \to [0, 1]$, both implemented as neural networks. The discriminator is trained to recognize "real" objects $\boldsymbol{x}$ taken from the data distribution $P_r$, while the generator is trained to map random latent vectors $\boldsymbol{z} \in \mathbb{R}^m$ to "fake" objects that fool the discriminator. Learning amounts to solving the minimax game:

$$\min_g \max_d f_{\text{GAN}}(g, d) \qquad f_{\text{GAN}}(g, d) := \mathbb{E}_{\boldsymbol{x} \sim P_r}[\log P_d(\boldsymbol{x})] + \mathbb{E}_{\boldsymbol{x} \sim P_g}[\log(1 - P_d(\boldsymbol{x}))] \qquad (1)$$

Here $P_g(\boldsymbol{x})$ is the distribution induced by the generator and $P_d(\boldsymbol{x}) := P_d(\text{real} \,|\, \boldsymbol{x})$ by the discriminator. It is worth noting that the value of the game defines (a constrained form of) the Jensen-Shannon entropy and that Eq. 1 can be viewed as divergence minimization (Nowozin et al., 2016; Mescheder et al., 2017). After training, new objects can be generated by sampling random vectors $\boldsymbol{z}$ and mapping them to object space with the generator $\boldsymbol{x} = g(\boldsymbol{z})$.

Goodfellow et al. (2014) have shown that, under idealized assumptions, the learned generator $g$ matches the data distribution, that is $P_g = P_r$. We report their theorem here for completeness:

**Theorem 1.** *So long as (i) $g$ and $d$ are non-parametric, and (ii) the leftmost expectation in Eq. 1 is approximated arbitrarily well by the examples alone, any global equilibrium $(g^*, d^*)$ of Eq. 1 satisfies $P_{d^*} \equiv \frac{1}{2}$ and $P_{g^*} \equiv P_r$.*

Finding Nash equilibria of non-convex games like Eq. 1 is notoriously hard, rendering GAN training very challenging. The most common optimization algorithm is alternating gradient descent, whereby $d$ and $g$ are optimized sequentially, or the more well-understood simultaneous gradient descent (Mescheder et al., 2017). A common failure state is mode collapse, whereby the generator outputs objects concentrated in a tiny portion of the object space. A plethora of theoretical and empirical remedies have been proposed, cf. (Salimans et al., 2016; Mescheder et al., 2018), including using alternative divergences (Nowozin et al., 2016; Arjovsky et al., 2017) and encouraging smoothness of the discriminator by, e.g., controlling the spectral norm of its parameters (Miyato et al., 2018). In our experiments, we apply some of these strategies to stabilize training.

In structured tasks, the objects of interest are usually discrete. GANs can be adapted to such settings by having the generator output a *categorical distribution* $\boldsymbol{\theta}(\boldsymbol{z})$ over $\mathcal{X}$ and sampling objects from the latter. Below, we will focus on stochastic generators of this kind, although alternatives do exist (Gulrajani et al., 2017). In this setting $P_g(\boldsymbol{x}) = \int P_g(\boldsymbol{x}|\boldsymbol{z})p(\boldsymbol{z})d\boldsymbol{z} = \int \boldsymbol{\theta}(\boldsymbol{z})p(\boldsymbol{z})d\boldsymbol{z} = \mathbb{E}_{\boldsymbol{z}}[\boldsymbol{\theta}(\boldsymbol{z})]$.

## 3 Generating Structures with CANs

Our goal is to learn to generate structures $\boldsymbol{x}$ consistent with respect to some validity constraint $\psi$ according to some unobserved distribution $P_r$. Throughout, we will make the following assumptions:

1. A single validity constraint $\psi$ is provided as input. This is without loss of generality: if multiple constraints are necessary, then $\psi$ can be taken to be their conjunction.

2. The constraint $\psi$ is compatible with the data distribution, that is, the support of $P_r$ falls entirely within the feasible region determined by $\psi$.

These assumptions hold in many tasks of interest, including all generative problems with known well-formedness requirements. For the sake of simplicity, we restrict our study to binary (i.e. 0–1) variables and logical constraints (aka formulas) only. This is a very general setup: any discrete structured space can be encoded using binary variables and formulas, at the cost of a larger model.

**Limitations of GANs**   Generators learned with the standard GAN training rule can output invalid structures, for two main reasons. First, for non-trivial constraints $\psi$, any finite set of examples that satisfy the constraint is insufficient to capture the full semantics of $\psi$. Second, in many cases of interest the examples are not even consistent with $\psi$. This more challenging case shows that regular GANs are easily lured into learning *not* to satisfy the constraint. More formally:

**Corollary 1.** *Under the assumptions of Theorem 1, given a target distribution $P_r$, a constraint $\psi$ consistent with it, and a dataset of examples $\boldsymbol{x}$ sampled i.i.d. from a corrupted distribution $\tilde{P}_r \neq P_r$ inconsistent with $\psi$, GANs associate non-zero mass to infeasible objects.*

Indeed, by Theorem 1, the optimal generator satisfies $P_g \equiv \tilde{P}_r$, which is inconsistent with $\psi$. Ergo, $\sum_{\boldsymbol{x}} \mathbb{1}\{\boldsymbol{x} \not\models \psi\} P_g(\boldsymbol{x}) > 0$. We stress that Theorem 1 captures the *intent* of GAN training, and thus this simple corollary shows that GANs are *by design* incapable of handling invalid examples.

**Constrained adversarial networks**   In order to avoid these issues, constrained adversarial networks (CANs) take both the examples and the validity constraint into account during learning. More specifically, the CAN value function is designed so that the generator maximizes the probability of generating valid structures $P_g(\psi) := \mathbb{E}_{\boldsymbol{x} \sim P_g}[\mathbb{1}\{\boldsymbol{x} \models \psi\}]$, that is:

$$f_{\text{CAN}}(g, d) := f_{\text{GAN}}(g, d) - \lambda \log P_g(\psi) \tag{2}$$

Here $\lambda > 0$ is a hyper-parameter controlling the importance of the constraint. Since the second term is always non-negative, the CAN value function upper bounds the GAN one (Eq. 1).

The second term is the so-called *semantic loss* (SL), proposed in (Xu et al., 2018) to inject knowledge into neural networks, and it is formally defined as $SL_\psi(g) \propto -\log P_g(\psi)$, where:

$$P_g(\psi) = \sum_{\boldsymbol{x}'} \mathbb{1}\{\boldsymbol{x}' \models \psi\} P_g(\boldsymbol{x}') = \mathbb{E}_{\boldsymbol{z}}[\sum_{\boldsymbol{x}' \in \mathcal{X} : \boldsymbol{x}' \models \psi} \prod_{i\,:\,x_i'=1} \theta_i(\boldsymbol{z}) \prod_{i\,:\,x_i'=0} (1 - \theta_i(\boldsymbol{z}))] \tag{3}$$

Recall that $\boldsymbol{\theta}(\boldsymbol{z})$ is the categorical distribution output by the generator $g$ for $\boldsymbol{z}$, while $\boldsymbol{x}' \models \psi$ means that $\boldsymbol{x}'$ satisfies constraint $\psi$. Hence the sum runs over all configurations $\boldsymbol{x}'$ consistent with $\psi$. The SL can also be viewed as the negative log-likelihood of $\psi$ w.r.t. the generator. This shows that, in Eq. 2, the SL rewards the generator $g$ proportionally to the mass it allocates to valid structures. Since the SL is the negative logarithm of a polynomial in $\boldsymbol{\theta}$, it is fully differentiable (so long as $P_\psi(g) \neq 0$, which is always the case in practice).

If the SL is given large enough weight, CANs are strongly encouraged to generate valid structures in expectation. Under the preconditions of Theorem 1, it can be shown that or $\lambda \to \infty$ CANs generate valid structures *only*:

**Proposition 1.** *Under the assumptions of Corollary 1, CANs associate zero mass to infeasible objects, irrespective of the discrepancy between $P_r$ and $\tilde{P}_r$.*

Intuitively, this holds because with $\lambda = \infty$ any global equilibrium $(g^*, d^*)$ of $\min_g \max_d f_{\text{CAN}}(g, d)$ *must* minimize the second term. If $g$ is non-parametric, then the minimum is attained with $\log P_{g^*}(\psi) = 0$, which is equivalent to $P_{g^*}(\psi) = 1$. This in turn implies that $P_{g^*}(\neg \psi) = 0$, proving the claim. Of course, as with standard GANs, the prerequisites are often violated in practice. Regardless, Proposition 1 works as a sanity check, and shows that – in contrast to GANs – CANs are appropriate for constrained generative tasks.

**Computing the semantic loss**   Naïve evaluation of the SL (Eq. 3) involves summing over all (exponentially many) possible configurations $\boldsymbol{x}'$, which is intractable. More generally, computing the SL amounts to weighted model counting, which is #P-complete (Chavira & Darwiche, 2008).

For many constraints of interest, however, the polynomial in Eq. 3 can be factorized into a much more compact form and evaluated very efficiently. Following Xu et al. (2018), we make use of

knowledge compilation to automatically factorize the polynomial by compiling it into an arithmetic circuit (more specifically, into a sentential decision diagram (SDD) (Darwiche, 2011)). When the resulting circuit is small enough, evaluation of the SL and of its gradient are extremely efficient, as shown in (Xu et al., 2018).

The main downside of knowledge compilation is that, depending on the complexity of the constraint $\psi$ at hand, the compiled circuit may be very large. This is less of a problem during training, which is often performed on powerful machines, but it can be an issue for inference – especially on embedded devices. However, in CANs the circuit is not required for inference (as it consists of a simple forward pass over the generator), and thus it can be thrown away after training. This means that CANs incur no space penalty during inference when compared to GANs.

**Dealing with intractable constraints** When fed a particularly complex constraint, knowledge compilation may produce a circuit too large even for the training stage. In this case, we approximate the semantic loss by first mapping the objects from $\mathcal{X}$ to an application-specific space where $\psi$ can be expressed in compact form, and then use the semantic loss on top of the transformed objects. We successfully employed this technique to synthesize mario levels where the goal tile is reachable from the starting tile; all details are provided below. The same technique can be exploited for dealing with very complex logical formulas beyond the reach of state-of-the-art knowledge compilation.

## 4 EXPERIMENTS

We implemented CANs using Tensorflow and used PySDD[1] to compile the constraint into a compact circuit for computing the Semantic Loss. We tested CANs using different generator architectures on one synthetic and two real-world structured generative tasks[2]. In all cases, we evaluated the objects generated by CANs and those of the baselines using three metrics (adopted from Samanta et al. (2018)):

- **validity** is the proportion of sampled objects that are valid;
- **novelty** is the proportion of valid sampled objects that are not present in the training data;
- **uniqueness** is the proportion of valid unique (non-repeated) sampled objects;

Our experimental evaluation aims at answering the following questions:

- **Q1** Can CANs with tractable constraints achieve better results than GANs?
- **Q2** Can CANs with intractable constraints achieve better results than GANs?
- **Q3** Can constraints be combined with rewards to achieve better results than using rewards only?

### 4.1 CONSTRAINED IMAGES

In this synthetic experiment, we used CANs to generate small, strongly constrained bitmap images. The training set is composed of $23,040$ black-and-white $20 \times 20$ images, each with a black background and two randomly placed random polygons – either a triangle ($30\%$ of the cases), a square ($30\%$), or a diamond ($40\%$). The two polygons are always different from each other, fully contained in the canvas, and do not overlap. The leftmost and rightmost columns are special, in that they contain parity information about the image. More specifically, row $r$ of the leftmost column encodes the parity bit for all pixels in the left half of row $r$, see Fig. 1 (left) for an illustration. The rightmost column does the same for the right half of the image. Fig. 1 (middle) shows a sample image. As a baseline GAN, we implemented the generator and discriminator as deconvolutional and convolutional networks, respectively, sampling from the concrete distribution (Maddison et al., 2016) and using the vanilla loss. The CAN uses the same architecture, except for the extra semantic loss term which encodes the parity constraints using XOR formulas.

---

[1]URL:https://pypi.org/project/PySDD/
[2]Details can be found in the Appendix. Our code is publicly available at the URL:ANONYMIZED

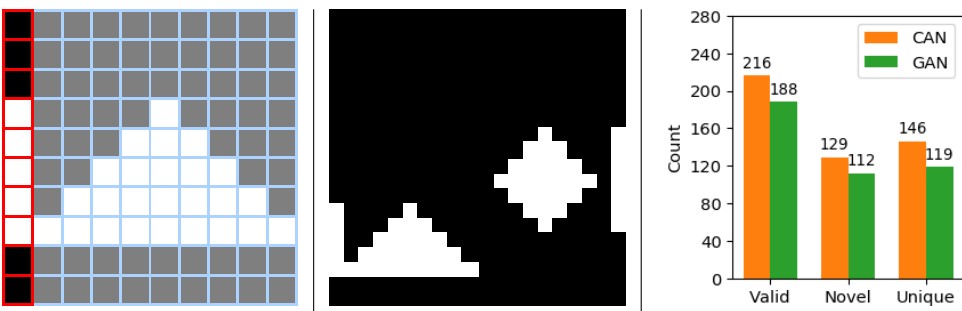

Figure 1: Left: illustration of the parity constraint at the left border. Middle: example from the synthetic dataset. Right: comparison of validity, novelty and uniqueness between CAN and GAN.

Fig. 1 (right) shows the results in terms of validity, novelty and uniqueness computed over 300 objects generated using CANs (with $\lambda = 1$) and Vanilla GANs, where novelty and uniqueness are computed over the subset of valid objects. CANs generate more valid objects, improving at the same time both novelty and uniqueness. These results allow to answer **Q1** affirmatively. Note that in this experiment, using CANs over Vanilla GANs results in a negligible training overhead. On average, training CANs takes 30 minutes against the 27 minutes taken by Vanilla GANs.

## 4.2 SUPER MARIO BROS LEVEL GENERATION

In the next experiment we show how CANs can help in the challenging task of learning to generate videogame levels from user-authored content. While procedural approaches to videogame level generation have successfully been used for decades, the application of machine learning techniques in the creation of (functional) content is a relatively new area of research (Summerville et al., 2018). On the one hand, modern video game levels are characterized by aesthetical features that cannot be formally encoded and thus are difficult to implement in a procedure, which motivates the use of ML techniques for task. On the other hand, the levels have often to satisfy a set of functional (hard) constraints that are easy to guarantee when the generator is hand-coded but pose challenges for current machine learning models.

In the following, we show how the semantic loss can be used to encode useful hard constraints in the context of videogame level generation. These constraints might be functional requirements that apply to every generated object or might be contextually used to steer the generation towards objects with certain properties. In our empirical analysis, we focus on *Super Mario Bros* (SMB), possibly one of the most studied video games in tile-based level generation. See Section 5 for related work in SMB level generation.

Recently, Volz et al. (2018) applied Wasserstein GANs (WGANs) (Arjovsky et al., 2017) to SMB level generation. The approach works by first training a generator in the usual way, then using an evolutionary algorithm called Covariance Matrix Adaptation Evolution Strategy (CMA-ES) to search for the best latent vectors according to a user-defined fitness function on the corresponding levels. We stress that this technique is orthogonal to CANs and the two can be combined together. Nonetheless, in the following we compare CANs with CMA-ES, as both techniques can be used to steer the network towards the generation of *playable* levels, i.e. having a feasible path[3] from the left-most to the right-most column of the level. Pictures in 2 shows two SMB levels generated by our CAN using the semantic loss.

In CMA-ES, the fitness function doesn't have to be differentiable and the playability can be computed on the output of an A* agent playing the level. Having the SL to steer the generation towards playable levels is not trivial, since it requires a differentiable definition of playability. Encoding the reachability of the right-most column with a propositional formula on the level variables $x$ is unfeasible (consider the size of the formula resulting from unrolling and grounding a compact first-order encoding). For this reason, our CAN applies the semantic loss to a different probability distribution $\theta$ over binary variables $r$, with the intended meaning that $r_i$ is true if and only if the tile $x_i$ is

---

[3]According to the game's physics.

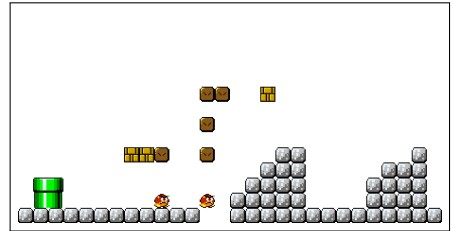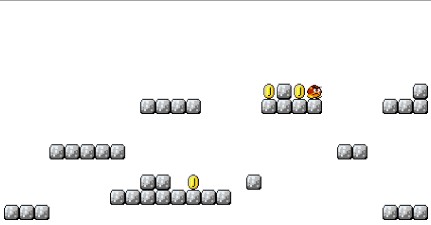

Figure 2: Illustration of two $14 \times 28$ tiles Super Mario Bros levels generated by CANs.

| Network type | Mario Level | Tested samples | Validity | Uniqueness | Novely | Training time | Inference time per sample |
|---|---|---|---|---|---|---|---|
| Baseline | mario-1-3 | 1000 | 9.80% | 100% | 100% | **1 h 15 min** | $\sim$ **40 ms** |
| MarioGAN + CMA-ES | mario-1-3 | 1000 | 65.90% | 100% | 100% | **1 h 15 min** | $\sim$ 22 min |
| CAN | mario-1-3 | 1000 | **71.60%** | 100% | 100% | 1 h 21 min | $\sim$ **40 ms** |
| Baseline | mario-3-3 | 1000 | 13.00% | 100% | 100% | **1 h 11 min** | $\sim$ **40 ms** |
| MarioGAN + CMA-ES | mario-3-3 | 1000 | **64.20%** | 100% | 100% | **1 h 11 min** | $\sim$ 22 min |
| CAN | mario-3-3 | 1000 | 62.30% | 100% | 100% | 1 h 16 min | $\sim$ **40 ms** |

Table 1: Results of using the semantic loss instead of post-processing with CMA-ES. Levels mario-1-3 and mario-3-3 has been chosen due to their higher complexity in being solved. The table shows the validity, uniqueness and novelty of samples after a testing session with A*. A valid sample is one that can be completely solved by the A* agent. The semantic loss has been activated from epoch 3000. Each run has lasted 5000 epochs with all the default hyper parameters defined in Volz et al. (2018)

reachable (in expectation) from the first column in the level $x$, with $x$ sampled from the generator's multinomial distribution $\gamma$ over tiles. Encoding the playability of a level is straightforward in $r$, although the problem becomes finding a fully differentiable transformation $P_R : \mathbf{\Gamma} \to \mathbf{\Theta}$.

We approximate $P_R$ with a feedforward neural network $\hat{P_R}$. Our approach consists in pretraining $\hat{P_R}$ with synthetic data $\{(\gamma_i, \hat{\theta}_i)\}$, consisting of syntetic distributions labelled using the tile-based pathfinder provided in Summerville et al. (2016). Inputs $\gamma_i$ are collected from previous runs of the generator, while $\hat{\theta}_i$ are computed by sampling 100 levels from $\gamma_i$ and averaging the reachability of each tile using the pathfinder's output. The data generation process is inexpensive, allowing to train $\hat{P_R}$ to reasonable performance for the task.

We adopt the same experimental setting, WGAN architecture and training procedure of Volz et al. (2018). The structured objects are $14 \times 28$ tile-based representations of SMB levels (e.g. Fig. 2) and the training data is obtained by sliding a 28 tiles window over levels from the *Video game level corpus* (Summerville et al., 2016). In this experiment we train on individual levels, as in (Volz et al., 2018), to be consistent with MarioGAN, but we stress that CANs can be trained on multiple levels.

Our semantic loss, applied on the output of the approximated reachability map $\hat{P_R}$, encodes that at least $4$ tiles of the right-most column are reachable. This choice is supported by two observations: if a tile is reachable, the same will probably be neighbours and the approximation given by $\hat{P_R}$ may satisfy too easily a constraint asking for only 1 reachable tile in the rightmost column.

Table 1 shows the playability, novelty and uniqueness of a batch of 1000 levels generated respectively by the CAN with a forward pass (with $\lambda = 0.01$, which validation experiments showed to be a reasonable trade-off between SL and discriminator loss) and by CMA-ES using the default parameters for the search. Novelty and Uniqueness are notably higher compared to the syntetic case due to a much larger output space[4]. Results show that CANs achieve comparable results on the first level, and substantial improvements on the second. Moreover, no significant negative effects can be seen on the generated material with respect to the quality of the ones generated by the original GAN.

Moreover, at the cost of pretraining $\hat{P_R}$ (1h35m on a 8-core machine with a GTX1080Ti), CANs avoid the execution of the A* agent during the generation, sampling high quality objects in milliseconds. On the other side, the training cost is slightly lower ($\sim 20\%$ less), but each run of CMA-ES, to find a single best individual, takes between 20 and 25 minutes on the previously described machine.

---

[4]The size of the output space is $13^{(14 \times 28)}$ solutions (with 13 being the number ofdifferent tiles)

| Reward for | Semantic loss | validity | uniqueness | diversity | QED | SA | logP |
|---|---|---|---|---|---|---|---|
| QED + SA + logP | False | 97.4 | 2.4 | 91.0 | 47.0 | 84.0 | 65.0 |
| | True | 96.59 (2.52) | 2.52 (0.25) | 98.81 (2.04) | 51.75 (1.55) | 90.74 (5.48) | 73.62 (1.14) |
| uniqueness | False | 99.19 (0.19) | 4.75 (2.35) | 66.87 (6.39) | 50.10 (1.68) | 37.26 (9.55) | 32.89 (4.55) |
| | True | 99.27 (0.16) | 17.41 (3.07) | 91.28 (2.87) | 41.54 (1.89) | 43.99 (8.38) | 33.94 (2.16) |

Table 2: Results of using the semantic loss on the MolGAN architecture. The diversity score is obtained by comparing sub-structures of generated samples against a random subset of the dataset. A lower score indicates a higher amount of repetitions between the generated samples and the dataset. The first row refers to the results reported in the MolGAN paper. Experiments were run 8 times each (rows 2, 3, 4) to obtain mean and std values, the latter in parentheses.

Combining the two approaches and refining the constraints for the generation of state-of-the-art Super Mario levels is a promising research direction and is left for future work.

With these results, we can answer **Q2** affirmatively.

### 4.3 MOLECULE GENERATION

In the next experiment, we test how effective is SL in conjunction with different forms of supervision on the task of generating molecules with certain desirable chemical properties. Specifically, we consider MolGANs (De Cao & Kipf, 2018), a model that combines the adversarial loss with a *reinforcement learning objective*, used to maximize the **druglikeness**, **sythesizability** and **solubility** of the generated molecules. The reward is computed by a network that is trained to match the score provided by an external cheminformatics software. In contrast with our previous experimental settings, here the structured objects are undirected graphs of bounded maximum size, represented by discrete tensors that encode the atom/node type (**padding atom** (no atom), **Carbon**, **Nitogren**, **Oxygen**, **Fluorine**) and the bound/edge type (**padding bond** (no bond), **single**, **double**, **triple** and **aromatic** bond).

As confirmed by both the results reported by the authors of MolGANs and our previous experiments, providing an additional loss term might perturb the already unstable adversarial game, possibly incurring in mode collapse. In the following, we explore the use of SL beyond the satisfaction of given constraints. Specifically, we observe that the formula can involve the latent variables too, and we show how this can be leveraged to increase the diversity of the generator's output and to mitigate the mode collapse. This can be achieved by using a portion of the latent vector (whose values are in the $[0, 1]$ domain) to trigger on and off SL terms that promote the presence of specific atoms in the molecule. Specifically, we apply the SL to MolGANs, making use of $4$ latent dimensions to control the presence of one of the $4$ types of atoms considered in the experiment. Each dimension represents the probability of having *at least one* atom of that type in the molecule, no matter the position. Assigning SL terms to specific subregion of the latent space is a general approach that can potentially be used to achieve some level of controllable constrained generation via the latent codes. We defer this research direction to future work.

In this experiment, we augment MolGAN with our constraints conditioned on the latent variables, starting from the first epoch. We consider two different variants of the reward network. In the first setting, the network implicitly rewards validity and the maximization of the three chemical properties at once: **QED** (druglikeness), **SA** (synthesizability) and **logP** (solubility). The experimental setting and evaluation metrics are identical to De Cao & Kipf (2018) except for the introduction of the SL, we thus report the same results for the baseline. In the second setting, the reward is proportional to the diversity of the generated batch, thus boosting the generator **uniqueness**. In this case, the training stops when the validity reaches $0.99$.

The results, as shown in Table 2, indicate that the SL term is boosting the diversity of the generated molecules without negatively affecting the other metrics with both reward functions. This preliminary results seem to suggest that CANs can be successfully coupled with a reinforcement learning objective, answering **Q3** affermatively. In this setting, using CANs produced a negligible overhead during the training with respect to the original model, providing further evidence that the technique doesn't heavily impact on the training.

## 5 RELATED WORK

Deep generative modeling has recently enjoyed substantial progress with the introduction of autoregressive models (Van Oord et al., 2016), variational autoencoders (VAEs) (Kingma & Welling, 2014; Rezende et al., 2014), flow-based approaches (Dinh et al., 2014), and GANs (Goodfellow et al., 2014). However, none of these approaches is designed to generate structures.

Traditional approaches to such tasks, like graphical models (Koller & Friedman, 2009; Richardson & Domingos, 2006) and probabilistic grammars (Talton et al., 2012), are ill-suited for complex tasks like image generation and do not support efficient inference under constraints. Tractable probabilistic circuits (e.g., probabilistic sentential decision diagrams (Kisa et al., 2014)) leverage knowledge compilation techniques – like CANs – and sport high capacity, efficient inference, and support for constraints, but they require the circuit even at inference time, meaning that inference can have large space requirements. Further, the runtime of inference depends on the size of the circuit. In contrast, inference in CANs boils down to a forward pass over the generator and does not rely on the circuit output by knowledge compilation. Finally, although solver-based approaches to constrained discrete sampling like (Ermon et al., 2013; Chakraborty et al., 2014) guarantee validity of the samples, they are also much slower than CANs and do not support learning the distribution from data.

Injecting knowledge into learned models, and neural networks in particular, is a long-standing aim of machine learning. Early approaches include grounding-specifica Markov Logic Networks (Lippi & Frasconi, 2009), which combine Markov logic with neural networks. More recent models include target-specific architectures (Rocktäschel & Riedel, 2017; Donadello et al., 2017), as well as frameworks based on probabilistic logics (Manhaeve et al., 2018) and T-norms (Marra et al., 2019). Other recent approaches include also constraint learning component (Wang et al., 2019; Sourek et al., 2018). However, none of these models is at the same time generative and probabilistic.

Posterior regularization (PR) has been used to introduce logic background knowledge into deep discriminative (Hu et al., 2016) and discriminative models (Hu et al., 2018). Like the Semantic Loss, PR can convert arbitrary discrete constraints into a loss. In addition, it does not require the target model to have an explicit density. The major downside of the PR is that it is not semantically consistent: rewriting a constraint (by applying e.g. De Morgan's laws) can lead to different loss functions. This never happens with the SL. Further, unlike the approach of (Hu et al., 2018), CANs do not need to define and train an approximate variational distribution: the SL is computed directly on the generator.

A number of machine learning techniques were applied to Super Mario Bros level generation, e.g. LSTMs (Summerville & Mateas, 2016), probabilistic graphical models (Guzdial & Riedl, 2016) and multi-dimensional Markov Chain Monte Carlo (Snodgrass & Ontanón, 2016). Torrado et al. (2019) leverages GANs, but differs from our work in that they consider level generation only and they constrain the mixture of tiles appearing in the level[5]. This technique cannot be easily generalized to arbitrary constraints. Many approaches to molecule generation use VAEs (Gómez-Bombarelli et al., 2018; Kusner et al., 2017; Dai et al., 2018). Closest to MolGANs are ORGANs (Guimaraes et al., 2017) and Sequence Tutor (Jaques et al., 2017), both using a reinforcement learning objective (with SeqGANs and RNNs respectively) to generate molecules as sequences in SMILE encoding. Finally, the approach of Seff et al. (2019) use MCMC to sample graph representations of valid molecules. Compared to CANs, their approach requires an expensive inference procedure based on Gibbs sampling, which must be additionally bootstrapped by providing a valid configuration. When the space is constrained by an arbitrary logical formula, as in our case, this last step is NP-complete.

## 6 CONCLUSION

We presented Constrained Adversarial Networks (CANs), a generalization of GANs in which the generator is encouraged *during training* to output valid structures. CANs make use of the semantic loss (Xu et al., 2018) to measure the mass allocated by the generator to invalid structures and penalize the latter accordingly. As in GANs, generating (likely) valid structures then amounts to a simple forward pass of the generator. Importantly, the data structures used by the SL – which can be large if the structural constraints are very complex – can be discarded after training. Our framework

---

[5]This work is a pre-print and the code is not publicly available yet, so no comparison could be made.

was proven to be effective in different structured generative tasks, improving the validity of the generated structures (on average) while keeping the computational cost of training under control and substantially reducing inference runtimes. We also showed how the constraints to be turned on and off at inference time, suggesting different uses for the SL, namely promoting diversity of the generator's outputs. Finally, our level generation experiment shows how to transform structures to a space in which the structural constraints are easier to encode and more compact.

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

# A    IMPLEMENTATION DETAILS

## A.1    CONSTRAINED IMAGES

The deep neural network for this experiment is based on a Deep Convolutional Generative Adversarial Networks (DCGANs) (Radford et al., 2015). Batch Normalization (BN) and ReLU are applied between the layers of the generator $G$. While Leaky ReLU with a slope of 0.2 has been used for the discriminator $D$. In the last layer of the generator we apply a gumbel-sigmoid activation function (GS).

For the training, Adam optimizer has been used with $\beta_1 = 0.5$ and $\beta_2 = 0.999$. The learning rate is constant throughout the training (250 epochs) and is equal to 0.0001. The batch size used during the experiments has been set to 64. Layers have been initialized using Xavier initialization for the generator and Orthogonal initialization for the discriminator. Finally, the size of the latent vector has been set to 64 and sampled from a normal distribution with $mean = 0.0$ and $std = 1.0$
Table 3 shows the network architecture.

| Part | Input Shape $\rightarrow$ Output Shape | Layer Type | Kernel | Stride |
|------|------|------|------|------|
|   | $(64) \rightarrow (1024)$ | Fully Conn. | - | - |
|   | $(1024) \rightarrow (3200)$ | Fully Conn. | - | - |
|   | $(3200) \rightarrow (\frac{h}{4}, \frac{w}{4}, 128)$ | Reshape |  |  |
| $G$ | $(\frac{h}{4}, \frac{w}{4}, 128) \rightarrow (\frac{h}{2}, \frac{w}{2}, 64)$ | Deconv. | $5 \times 5$ | $2 \times 2$ |
|   | $(\frac{h}{2}, \frac{w}{2}, 64) \rightarrow (h, w, 1)$ | Deconv. | $5 \times 5$ | $2 \times 2$ |
|   | $(h, w, 1) \rightarrow (\frac{h}{2}, \frac{w}{2}, 64)$ | Conv. | $5 \times 5$ | $2 \times 2$ |
|   | $(\frac{h}{2}, \frac{w}{2}, 64) \rightarrow (\frac{h}{4}, \frac{w}{4}, 128)$ | Conv. | $5 \times 5$ | $2 \times 2$ |
| $D$ | $(\frac{h}{4}, \frac{w}{4}, 128) \rightarrow (3200)$ | Reshape |  |  |
|   | $(3200) \rightarrow (1024)$ | Fully Conn. | - | - |
|   | $(1024) \rightarrow (32)$ | Fully Conn. | - | - |
|   | $(32) \rightarrow (1)$ | Fully Conn. | - | - |

Table 3: Constrained Images network architecture.

## A.2    SUPER MARIO BROS LEVEL GENERATION

The deep neural network for this experiment is based on the DCGANs used in (Volz et al., 2018). Batch Normalization (BN) and ReLU are applied between the layers of the generator $G$. While Leaky ReLU with a slope of 0.2 has been used for the discriminator $D$. In the last layer of the generator we apply a Softmax activation function to obtain probabilities that are finally given as inputs to the Semantic Loss. On the other hand, the generation of samples is done through the application, always on the generator, of a stretched Softmax function followed by a Tensorflow OneHotCategorical layer, which allows to samples levels. In our opinion, this is a best way of generating objects with respect to the simple *argmax* applied directly on the generator by (Volz et al., 2018).

The networks have been trained using the WGAN guidelines (Arjovsky et al., 2017). Thus, the number of iterations on the discriminator has been set to 5 for each iteration on the generator. RMSProp has been used as loss optimizer, with the learning rate constant throughout the training (10000 epochs) and equal to 0.00005. The batch size used during the experiments has been set to 32. Layers have been initialized using Normal initializer for both the generator and the discriminator. Moreover, weight clipping is applied on the layers of $D$ with $c$ equals to 0.01. Finally, the size of the latent vector has been set to 64 and sampled from a normal distribution $\mathcal{N}(0, 1)$
Table 4 shows the network architecture (as said, each layer is followed by batch normalization and by a ReLU/Leaky ReLU activation function).

The feedforward neural network $\hat{P}_R$ has been implemented using a CNN with the architecture in Table 5.

| Part | Input Shape $\rightarrow$ Output Shape | Layer Type | Kernel | Stride |
|------|----------------------------------------|------------|--------|--------|
|   | $(32) \rightarrow (1, 1, 32)$ | Reshape. | - | - |
|   | $(1, 1, 32) \rightarrow (\frac{h}{8}, \frac{w}{8}, 16)$ | Deconv. | $4 \times 4$ | $1 \times 1$ |
| $G$ | $(\frac{h}{8}, \frac{w}{8}, 16) \rightarrow (\frac{h}{4}, \frac{w}{4}, 8)$ | Deconv. | $4 \times 4$ | $2 \times 2$ |
|   | $(\frac{h}{4}, \frac{w}{4}, 8) \rightarrow (\frac{h}{2}, \frac{w}{2}, 4)$ | Deconv. | $4 \times 4$ | $2 \times 2$ |
|   | $(\frac{h}{2}, \frac{w}{2}, 4) \rightarrow (h, w, 13)$ | Deconv. | $4 \times 4$ | $2 \times 2$ |
|   | $(h, w, 13) \rightarrow (\frac{h}{2}, \frac{w}{2}, 64)$ | Conv. | $4 \times 4$ | $2 \times 2$ |
|   | $(\frac{h}{2}, \frac{w}{2}, 64) \rightarrow (\frac{h}{4}, \frac{w}{4}, 128)$ | Conv. | $4 \times 4$ | $2 \times 2$ |
| $D$ | $(\frac{h}{4}, \frac{w}{4}, 128) \rightarrow (\frac{h}{8}, \frac{w}{8}, 256)$ | Conv. | $4 \times 4$ | $2 \times 2$ |
|   | $(\frac{h}{8}, \frac{w}{8}, 256) \rightarrow (1, 1, 1)$ | Conv. | $4 \times 4$ | $1 \times 1$ |

Table 4: Super Mario Bros Level Generation network architecture.

| Input Shape $\rightarrow$ Output Shape | Layer Type | Kernel | Stride |
|-----------------------------------------|------------|--------|--------|
| $(h, w, 13) \rightarrow (h, w, 8)$ | Conv. | $3 \times 3$ | $1 \times 1$ |
| $(h, w, 8) \rightarrow (h, w, 16)$ | Conv. | $5 \times 5$ | $1 \times 1$ |
| $(h, w, 16) \rightarrow (h, w, 24)$ | Conv. | $7 \times 7$ | $1 \times 1$ |
| $(h, w, 24) \rightarrow (h, w, 32)$ | Conv. | $9 \times 9$ | $1 \times 1$ |
| $(h, w, 32) \rightarrow (h, w, 64)$ | Conv. | $11 \times 11$ | $1 \times 1$ |
| $(h, w, 64) \rightarrow (h, w, 96)$ | Conv. | $13 \times 13$ | $1 \times 1$ |
| $(h, w, 96) \rightarrow (h, w, 128)$ | Conv. | $15 \times 15$ | $1 \times 1$ |
| $(h, w, 128) \rightarrow (h, w, 192)$ | Conv. | $17 \times 17$ | $1 \times 1$ |
| $(h, w, 192) \rightarrow (h, w, 32)$ | Fully Conv. | - | - |
| $(h, w, 32) \rightarrow (h, w, 2)$ | Fully Conv. | - | - |

Table 5: Reachability Network architecture.

### A.3 MOLECULE GENERATION

The MolGAN architecture is composed of three networks, the generator $G$, the discriminator $D$ and the reward network $R$. $D$ and $R$ share the same architecture, but are trained with different objectives. $R$ must estimate either the product of the QED, SA, logP metrics, or the uniqueness of a batch of samples. All these metrics are in $[0, 1]$ so that $R$ should output $1$ given a perfect sample (w.r.t. the aforementioned metrics) or a batch made of different samples. $G$ is optimized to produce samples that maximize the output of $R$ and are convincing to $D$, on top of that, the semantic loss is also applied.

While $R$ is trained in parallel with $D$ and $G$, the loss of $G$ with respect to the output of $R$ is activated only after 150 epochs, at which point the adversarial loss stops being used. The semantic loss is applied to $G$ from the start to the end of the training. The weight of the semantic loss is equal to $0.9$, whereas the adversarial loss of $G$ has a weight of $0.1$.

Improved WGAN is used as the adversarial loss between $G$ and $D$ ($n_{critic} = 5$), whereas $R$ is trained to estimate the desired target by mean squared error, and $G$ is trained with deep deterministic policy gradient w.r.t. the output of $R$, which is seen as a reward to maximize.

Training is ongoing until a stop condition is met, either the uniqueness of the batch falling below $0.2$ if $R$ is an estimator of the three chemical properties, or the validity reaching $0.99$ if $R$ should output the uniqueness of the batch. During training, the learning rate is set at a constant value of $0.001$, the batch size is 32 and there is no dropout. Adam with $beta1 = 0.9$ and $beta2 = 0.999$ is the optimizer of choice. Batch discrimination is used.

Results are obtained by evaluating a batch of 5000 generated samples.

The input noise $z$ has dimension 32 if the semantic loss is not applied, 36 if it is applied; with the first 32 dimensions sampled from a standard normal distribution and the last four from a uniform distribution in $[0, 1]$.

The maximum number of nodes for each molecule is 9, with 5 possible atom types and 5 bond types. Each molecule is represented by two matrices, one mapping each node to a label, and one adjacency matrix informing about the presence or lack of edges between nodes, and their type.

The input noise is received by $G$ and processed by three fully connected layers of $128, 256, 512$ units each, while tanh acts as the activation function; a linear projection followed by a softmax is

then applied to the output of the last layer to have it matching the size of the adjacency matrices. $D$ and $R$ share the same architecture (no parameters are shared), based on two relational graph convolutions (De Cao & Kipf, 2018) of $64, 32$ hidden units, followed by an aggregation as in (De Cao & Kipf, 2018) to obtain a graph level representation of 128 features. Two fully connected layers of $128, 1$ units then reduce the graph embedding to a single output value, with $\tanh$ as the activation for the hidden layer and with sigmoid being applied on the output in the case of $R$.

