# OpenReview forum: "Efficient generation of structured objects with Constrained Adversarial Networks"
_ICLR.cc/2020/Conference — Reject_

### Official Review · AnonReviewer3 · 2019-10-21
**Official Blind Review #3**

**Rating:** 3

**Review:**

In this paper the authors present a Generative Adversarial Neural Networks with Xu et al.’s semantic loss applied to the generator. They call this GAN a Constrained Adversarial Network or (CAN) and identify it as a new class of GAN. The authors present three different problem domains for their experiments focused on the generation of constrained images, chunks of Super Mario Bros.-style levels, and molecules. For each domain they include particular constraints for the semantic loss, which biases the generator towards creating valid content according to these constraints.

The paper at present has a number of issues holding it back. First, I am not convinced by the author’s claims that the application of an existing loss function to the generator is sufficient to identify a new class of GAN. Second, there is a lack of technical detail in the experiments necessary to replicate them. Third, there is a lack of discussion of the experimental results to place them in context for readers. Finally following from the earlier points, there seems to be a lack of technical contributions in the paper.

I certainly agree with the authors about the inability of GANs to learn structural constraints with insufficient training data, as this has been demonstrated in many examples of prior work. I also agree that particular problem domains, as identified by the authors, have stronger structural requirements. However, it is unclear to me why in these instances one would use GANs and not some alternative approach such as constraint-based solvers. Or even if one wanted to employ GANs, what the benefit of adapting the constraints into a loss function is compared to say constraining generated output in a post-hoc process.

The descriptions of the two of the three experiments do not include any discussion of the GAN architectures or hyperparameters. While this is not strictly necessary in the paper text some discussion in an appendix or a citation to a prior application of the architecture(s) would be appropriate. Without this, it is impossible for future researchers to replicate these results. Further, it is difficult for readers to place the results in context. For individual experiments, such as the Super Mario Bros. experiment, it is unclear why certain choices were made. For example, why train a GAN on just level 1-3 or 3-3, and not train a single model on multiple levels as is common in the field of procedural content generation via machine learning.

There is a lack of discussion in the paper on the results of each experiment. For example, the output of the GANs for all the experiments seems quite low, and the differences in terms of the results between the GAN and the CAN across the experiments do not seem to be substantial. Some discussion to put this into context for readers would be helpful.

As far as I can understand the primary technical contributions of the paper are: (1) the application of Xu et al.’s semantic loss to GANS, (2) the constraints developed for the three experiments, and (3) the results of the three experiments. I am unconvinced of the utility of these contributions to a general machine learning audience.

---

Updated my review as the authors included extra detail regarding the experiments in a new draft, which helped with the reproducibility issue. However, I am still unconvinced in the contributions of the paper outside of what I previously listed. While I am also unfamiliar with any prior example demonstrating that GANs produce invalid structure, this is not a surprising result. Especially as validity can be defined in an arbitrary, domain-specific manner.


**Experience Assessment:**

I have published in this field for several years.

**Review Assessment: Checking Correctness Of Derivations And Theory:**

N/A

**Review Assessment: Checking Correctness Of Experiments:**

I assessed the sensibility of the experiments.

**Review Assessment: Thoroughness In Paper Reading:**

I read the paper thoroughly.

---

> ### Author Response · Authors · 2019-11-07
> **Reply to Reviewer #3**
>
> Thank you for your careful review.
>
>
> Naming: We only coined the term CAN for recognizability.  We are open to changing the
> name of our approach, if necessary.
>
>
> Missing technical details and discussion: We will provide any missing material directly in an updated version of the manuscript very soon.
>
>
> Technical contributions: Our paper does not equate to GAN + SL. Our technical contributions are:
>
> Showing that GANs by design allocate non-zero mass to invalid structures  whenever the dataset is noisy.
>
> Fixing this issue by pairing the generator with the SL and showing that the resulting architecture provably produces valid structures only (in the limit of $\lambda \to \infty$). Compared to alternative architectures, the resulting model enjoys exact probabilistic semantics and can natively handle any arbitrary discrete constraint without special-purpose architectural modifications.
>
> We show that the SL can be successfully supplemented with a neural component in practice when the constraints are beyond the reach of model counting technology.
>
> We show that constraints can be turned on and off at test time despite being "baked" into the generator at training time.
>
> Using alternatives like constraint solvers: State-of-the-art approaches to discrete (weighted) sampling under constraints are either solver-based [a,b] or knowledge-compilation-based [c].  Approaches in the first group are approximate and rely on invoking a (usually NP-hard) oracle.  Also, these approaches tackle sampling, not learning.  Approaches in the second group, like PSDDs [c], make use of the same knowledge compilation techniques that underlie the Semantic Loss.  If the constraints are very complex, KC may output very large circuits (polynomials) that in turn seriously affect inference runtime and space requirements.  CANs on the other end only need the circuit during training (which can be handled on larger machines and is only performed once).  Further, the complexity of inference in CANs does not depend on the complexity of the constraints, while in PSDDs it does.  Finally, PSDDs can be learned from data, but just like GANs, they cannot acquire and apply constraints from a handful of potentially noisy examples.
>
> [a] Chakraborty et al. “Distribution-Aware Sampling and Weighted Model Counting for SAT”, 2014.
> [b] Ermon et al. “Embed and Project:Discrete Sampling with Universal Hashing”, 2013.
> [c] Kisa et al. “Probabilistic sentential decision diagrams”, 2014.
>
> We will make sure to discuss discrete sampling technology more in detail in the related work.
>
>
> CESAGAN: The main differences with CANs are as follows:
>
> (1) It is unclear if CESAGANs (and specifically count vectors plus an embedding layer) can be extended to deal with arbitrary logical constraints, like CANs do.
>
> (2) In CESAGANS, the count vector is given as input to the discriminator, not directly to the generator, which introduces one layer of indirection;  in CANs this is not necessary.
>
> (3) The relationship between the count vector and the decision of the discriminator must be learned, which is non-trival without extra supervision and, again, more indirect than imposing the SL loss term in CANs.
>
> (4) As in MarioGANs, the supervision on the playability is given by an A* agent, resulting in a much computationally expensive training.  In our experiments the agent is only used for performance evaluation.
>
> (5) CESAGANs focus on level generation and were not tested on other generative tasks, while we applied CANs to multiple applications.
>
> An empirical comparison could be interesting, but [d] is not peer-reviewed and the code is not available.
>
> Finally, the pre-print [d] was uploaded to ArXiV after the ICLR ‘20 deadline.
>
> [d] https://arxiv.org/abs/1910.01603
>
>
> Missing details and missing discussion: Thank you for pointing out this deficiency of our paper.  We will upload an updated version soon.
>
>
> Training on a single level: We can definitely use more than one level during the training. In order to compare with MarioGAN, we trained the generation on a single level. In the MarioGAN paper, the authors use 1-1.  We choose 1-3 and 3-3 as they are more challenging with respect to the playability. This was mentioned in the caption of Table 1; we will make sure to make this more prominent.

---

### Official Review · AnonReviewer2 · 2019-10-22
**Official Blind Review #2**

**Rating:** 3

**Review:**

This paper proposed Constrained Adversarial Networks (CAN), which incorporates structural constraints by augmenting a penalty term in the training object. The penalty term is formulated as the semantic loss proposed in [1] which can handle any logical constraints. Experiments are demonstrated to show the advantage of CAN over standard GAN in terms of whether the generated samples satisfy the hard constraints, and whether they are novel and unique.

First, I'd like to thank the authors for making this paper easy to follow. I like the idea of encouraging constraints for generative models, which is useful and interesting. However, given the published paper [1], this work seems to be a bit incremental.

The semantic loss for incorporating constraints and the knowledge compilation techniques for efficient evaluation are both introduced and discussed in [1]. The novelty of this paper is to apply these techniques to generative models, which seem to be a bit straightforward. A similar idea is proposed in [2], where the authors also discussed logical constraints and generative models, but they call the augmented penalty as 'posterior regularization'. I will be interested in a comparison to their method in terms of both methodology level and experiment level.

Overall the contribution of this paper does not seem to be strong enough. I would personally vote for weak rejection.

[1] Xu, Jingyi, et al. "A semantic loss function for deep learning with symbolic knowledge." arXiv preprint arXiv:1711.11157 (2017).
[2] Hu, Zhiting, et al. "Deep generative models with learnable knowledge constraints." Advances in Neural Information Processing Systems. 2018.

**Experience Assessment:**

I have read many papers in this area.

**Review Assessment: Checking Correctness Of Derivations And Theory:**

I did not assess the derivations or theory.

**Review Assessment: Checking Correctness Of Experiments:**

I assessed the sensibility of the experiments.

**Review Assessment: Thoroughness In Paper Reading:**

I read the paper at least twice and used my best judgement in assessing the paper.

---

> ### Author Response · Authors · 2019-11-07
> **Replies to Reviewer #2**
>
> Thank you for your thoughtful review.
>
>
> Incrementality wrt [1]: Our contributions go beyond applying the SL to GANs:
>
> We show that GANs by design generate invalid structures if the data is noisy.  To the best of our knowledge, previous papers on deep generative models for structured outputs do not look into this at all.
>
> CANs generalize beyond existing ad-hoc architectures, as thanks to the SL they can natively handle any arbitrary discrete constraint.
>
> We discuss one case where the SL *cannot* be used as-is (i.e., for the level-wide reachability constraint in the mario experiment) and show that in practice it is possible to replace parts of the SL using a neural network.
>
> We show that the constraints, although "baked" into the generator at training time, can be turned on and off using an InfoGAN-like approach (cf. the molecules generation experiment). This technique can also be used to sample valid objects from different modes, thus also mitigating mode collapse.
>
> We agree that these contributions were not made clear enough, and we will definitely amend the paper to this effect.
>
>
> Comparison with [2]: We had initially considered using posterior regularization for CANs, but a major issue it is that rewriting the constraint (e.g. applying De Morgan) is not guaranteed to preserve the semantics and thus may change the loss function.  This issue does not affect the SL.  Moreover, since the SL can be evaluated efficiently in the GAN case, we have no need to fit a variational distribution $q$.  We agree that an empirical comparison with [2] is in order, however their code is not publicly available.
>
>
> Please expect an updated manuscript shortly.

---

### Official Review · AnonReviewer1 · 2019-10-22
**Official Blind Review #1**

**Rating:** 3

**Review:**

The authors describe a method to improve the performance of generative adversarial networks in the task of generating structured objectives that have to satisfy complicated constraints. The proposed solution involves using an additional term in the GAN objective that penalizes the generation of invalid samples. This term, called the semantic loss, is given by a multiple of the log probability of the model generating valid samples.

Clarity:

The paper is not very well written and several parts need to be clarified. In particular, in equation 3. What is theta in this equation?  how is it obtained? The authors mention briefly how their method could be used to deal with intractable constraints, but they're almost no specific details or examples of how this is done in practice. The proposed approach relays on the knowledge compilation method, but they're very few details of it in the document. Is it used at all in the experiments?

I am concern about the lack of reproducibility of the paper. I believe, from the paper as it is, it will be impossible to reproduce the results. There are no details about public code release, hyper-parameters settings, etc. For example,
in section 4.3 the authors mention that they condition the constraint on 5 latent dimensions without giving details about which dimensions exactly.

Significance:

It is hard to quantify the significance of the contribution. The constrained images problem is very toy and simple and the experiments with molecules do not include any baseline (only the GAN model without the constraint). There have been
many recent contributions improving the validity of generative models for molecules and the authors do not compare with any of them.

The authors also fail to cite relevant work such as

Jaques, Natasha, et al. "Sequence tutor: Conservative fine-tuning of sequence
generation models with kl-control." Proceedings of the 34th International
Conference on Machine Learning-Volume 70. JMLR. org, 2017.

Seff, Ari, et al. "Discrete Object Generation with Reversible Inductive
Construction." arXiv preprint arXiv:1907.08268 (2019).

Novelty:

The proposed approach is rather incremental and lacks novelty. It consists in just applying the semantic loss approach of Xu et al. 2018 to GAN training, with very limited new methodological or algorithm contributions.

Quality:

The experiments performed are not strong enough to validate the proposed method. The authors do not consider strong baselines in their evaluations.

Summary:

I find that the problem addressed by the authors is highly relevant and the proposed approach has the potential to be useful in practice. However, the paper needs to be improved regarding its clarity, reproducibility and strength of experiments before it can be accepted for publication.

**Experience Assessment:**

I have published in this field for several years.

**Review Assessment: Checking Correctness Of Derivations And Theory:**

I assessed the sensibility of the derivations and theory.

**Review Assessment: Checking Correctness Of Experiments:**

I assessed the sensibility of the experiments.

**Review Assessment: Thoroughness In Paper Reading:**

I read the paper at least twice and used my best judgement in assessing the paper.

---

> ### Author Response · Authors · 2019-11-07
> **Answer to Reviewer #1**
>
> Thank you for your detailed review.
>
>
> Clarity: We are happy to rewrite any parts of the paper that may be unclear.
>
> Eq. 3: Theta is the distribution output by the stochastic generator, as explained at the end of the section on GANs.  We will make it more clear.
>
>
> No details on knowledge compilation: All of the relevant details can be found in Xu et al.  We will update the short introduction to KC in the methods section to be more self-contained.
>
> Is KC used in the experiments? All constraints used in the experiments required us to apply KC.  We compiled the constraints into SDDs using the pysdd library.
>
> Some constraints result in extremely large circuits that cannot fit in memory. We show how to deal with these overly complex constraints  by using simpler constraints on a projected space. We remark that this is not exclusive of our approach but, to the best of the authors’ knowledge, this is the first work that applies these ideas to effectively approximate the exact SL. We mentioned in  Sections 1,3 that this approach is used in the level generation task to approximate with a propositional formula the reachability (cf. pages 5-6). Although showcased in the level generation setting only, the approach is general. We will clarify these aspects in the revised version.
>
>
> Reproducibility: Reproducibility is crucial for us. We will shortly share an archive with the anonymized version of the code with the reviewers.  We will also add a link to the code to the paper, as well as details on the architectures and hyperparameters in an Appendix.  An updated manuscript will be uploaded soon.
>
>
> Significance: The focus of the molecules generation experiment is not on comparing the SL with other forms of supervision nor on comparing every existing approach in molecule generation, but rather on showing how the SL can be used in conjunction with reinforcement-based approaches (as used in MolGAN, ORGANs, etc.) to mitigate the mode collapse and foster diversity.  This is achieved by applying different constraints on different subregions of the latent space.
>
> Our experiments show that the SL improves the quality of the generated structures when combined with constrained baselines.
>
> Notice that in some cases the baseline already generates mostly valid structures, as in the molecule generation experiment.  In this case, there is little gain in trying to improve the validity further.  For this reason, we use the SL to improve the other quality measures.  The results show that indeed the SL improves uniqueness and diversity, see Table 2.
>
>
> Baselines: Thank you for the additional references.
>
>
> Technical contributions: We stress that our contribution does not equate to combining GANs and the SL.  Our contributions are as follows:
>
> We prove that GANs by design allocate non-zero mass to invalid structures whenever the dataset is noisy.
>
> We fix this issue by pairing the generator with the SL and showing that the resulting architecture provably produces valid structures only (in the limit of $\lambda \to \infty$). Compared to alternative architectures, the resulting model enjoys exact probabilistic semantics and can natively handle any arbitrary discrete constraint without special-purpose architectural modifications.
>
> We show that the SL can be successfully supplemented with a neural component in practice when the constraints are beyond the reach of model counting technology.
>
> We show that constraints can be turned on and off at test time despite being "baked" into the generator at training time.

---

### Decision · Program_Chairs · 2019-12-19

**Decision:**

Reject

**Comment:**

This paper develops ideas for enabling the data generation with GANs in the presence of structured constraints on the data manifold. This problem is interesting and quite relevant to the ICLR community. The reviewers raised concerns about the similarity to prior work (Xu et al '17), and missing comparisons to previous approaches that study this problem (e.g. Hu et al '18) that make it difficult to judge the significance of the work. Overall, the paper is slightly below the bar for acceptance.